# Does the quality of pain relief after major surgery influence the risk of postoperative complications? A prospective observational study

Christine Kubulus[1,2]*, Marcus Komann[3], Markus Paxian[4], Ann-Kristin Schubert[5], Daniel Schwarzkopf[3], Norman Rose[6], Winfried Meissner[3], Ursula Marschall[7], Johannes Dreiling[3], Carolin Fleischmann-Struzek[6], Thomas Volk[1,2]*, the net-ra investigators¶

1 Faculty of Medicine, Anaesthesiology, Saarland University, Homburg, Germany, 2 Outcomes Research Consortium˚, Houston, Texas, United States of America, 3 Department of Anaesthesiology and Intensive Care Medicine, Friedrich Schiller University Jena, Jena University Hospital, Jena, Germany, 4 Department of Anaesthesiology, Emergency Medicine, Intensive Care and Pain Therapy, Ubbo-Emmius-Hospital, Aurich, Germany, 5 Department of Anesthesiology and Intensive Care Medicine, University Hospital Giessen and Marburg, Campus Marburg, Philipps-University Marburg, Marburg, Germany, 6 Institute of Infectious Diseases and Infection Control, Friedrich Schiller University Jena, Jena University Hospital, Jena, Germany, 7 Department of Medicine and Health Services Research, BARMER Health Insurance, Wuppertal, Germany

¶ Membership of the net-ra investigators is listed in the Acknowledgments.
* Christine.Kubulus@uks.eu (CK); Thomas.Volk@uks.eu (TV)

## Abstract

### Objectives

Effective postoperative acute pain management continues to be a challenge. It remains uncertain whether poorly controlled postoperative pain influences the risk of postoperative complications. Therefore, we aimed to investigate whether indicators of poor pain control increase the likelihood of cardiac, pulmonary, infectious, thrombo-embolic, and surgical complications, as well as of prolonged use of analgesics.

### Methods

This prospective observational study combines treatment data from the German net-ra registry and claims data from the second-largest public health insurer BARMER (Mar 1, 2021-Mar 31, 2022). A total of 539 adult inpatients who had undergone major surgery and received planned postoperative care from acute pain services were analyzed. Adjusted binary logistic regression models were fitted to compare patients with inadequately (NRS > 3) and adequately controlled pain, with (NRS > 6) and without pain peaks, and with slow or rapid pain recovery (median split of the time to sustained adequate pain control NRS ≤ 3) with regard to the risk of postoperative complications and prolonged use of analgesics as a proxy of chronic postoperative pain.

**Data availability statement:** The data cannot be made publicly available because of ethical and legal regulations. The data is available to researchers who meet the criteria for access to confidential data, upon reasonable request and after approval by the responsible data protection authorities and the responsible ethics committee, via the organizational committee of the net-ra registry (kontakt@net-ra.eu).

**Funding:** The LOPSTER project on which this publication is based was funded by the German Innovation Committee of the Federal Joint Committee (G-BA), Berlin, Germany under grant no. 01VSF19019 to DS. The funders had no role in study design, data collection and analysis, decision to publish, or preparation of the manuscript.

**Competing interests:** WM reports institutional funding from the European Commission, Gemeinsamer Bundesausschuß (GBA), Medtronic, Pfizer, Mundipharma, Grünenthal, and Vertanical as well as personal fees from Merck, Sanofi, MSD, Tafalgie, Kyowa, Mundipharma, Grünenthal and Ethypharm. TV reports institutional funding from Sedana medical, Ratiopharm, Pfizer, InfectoPharm, and Cyto Sorbents Europe as well as lecture fees from Pajunk and CSL Behring. TV is the past present of the European Society of Regional Anaesthesia and Pain Therapy. CK, MK, JD, CFS, MP, AS, DS, NR, and UM report no conflicts of interests.

## Results

Patients with inadequately controlled pain within the first three postoperative days had more than twice the risk of complications (adjOR 2.56; 95% CI 1.43–4.80, p = 0.002), as did patients with slow pain recovery (adjOR 2.21; 95% CI 1.35–3.64, p = 0.002). No significant effect could be observed for pain peaks (adjOR 1.27; 95% CI 0.64 to 2.42, P = 0.478). Inadequate pain control did not significantly affect prolonged use of analgesics (adjOR 1.87; 95% CI 0.98–3.72, p = 0.064), nor did pain peaks or recovery speed show any influence.

## Discussion

We observed a clear link between postoperative quality of pain control and complications, along with a trend towards prolonged use of analgesics. Therefore, postoperative acute pain should be regularly assessed and minimized until resolved. Further research into patient- and procedure-specific factors is essential to reduce adverse pain-related outcomes.

---

## Introduction

Currently, there is a wide range of options for pain relief after surgery, but many patients still experience moderate to severe acute postoperative pain [1,2]. Poorly controlled pain, which may occur after surgical trauma, triggers a variety of stress-related physical reactions [3,4]. The influence of these reactions on the cardiovascular, coagulation, and immune systems is well known [5], and they may contribute to postoperative cardiac, pulmonary, infectious, thromboembolic, and surgical complications. If complications occur in the immediate postoperative or long-term course, they are often regarded as fateful and not as a possible consequence of inadequate pain treatment.

However, there are only a few studies on the effects of postoperative acute pain on postoperative complications. Greater pain intensity is associated with a higher risk of myocardial injury [6] and of postoperative infectious and non-infectious complications [7,8]. In addition, the intensity of postoperative acute pain is associated with an increased tendency towards pain chronification [2]. Depending on the surgical procedure, the incidence of postoperative chronic pain is estimated between 10–50% and represents a major clinical and health economic problem [9].

We therefore hypothesize that adequate pain control in hospitalized adult patients undergoing major surgery influences the risk of postoperative complications. Accordingly, we tested the following hypotheses: (a) postoperative complications occur more frequently in patients with poor pain control regarding both severity and duration, (b) the risk of pain chronification, reflected in prolonged postoperative analgesic use, is higher in patients with poor pain control.

## Materials and methods

This study is part of the LOPSTER (long-term outcome of perioperative pain therapy based on routine data) project, which was funded by the German Innovation

Committee of the Federal Joint Committee (G-BA, Berlin, Germany). Ethical approval for this study was provided by the Ethical Committee of Friedrich Schiller University, Jena, Germany (Chairperson Prof. Dr. U. Brandl) on October 29, 2020 (identification number: 2020–1952-Daten). Written informed consent was a prerequisite for participation in the study. This manuscript adheres to the RECORD guideline [10].

## Study design

This prospective, multicenter, observational cohort study analysis using routinely collected data was designed to compare the risk of postoperative complications and prolonged analgesic use in adult patients with adequate versus inadequate pain management after major surgery. Data on postoperative acute pain therapy of patients who consented to participate in the study were obtained from the German Network for Safety in Regional Anaesthesia and Acute Pain Therapy (net-ra) registry. The registry was established in 2007 by the German Society for Anaesthesiology and Intensive Care Medicine and the Professional Association of German Anaesthetists. In addition to pre-, intra- and postoperative data on regional anesthesia procedures, it contains detailed information on postoperative pain therapies performed by acute pain services at German hospital centers using a standardised protocol. Acute pain services are generally used when severe pain is expected and basic analgesia is not considered sufficient. The prospective study design allowed for training of the participating centers prior to patient enrollment, ensuring that routinely collected data were as complete and accurate as possible and that inter-center variability was minimized. Pain values at rest and during movement were recorded using a standardized procedure on an 11-point numerical rating scale (NRS) (0 = no pain, 10 = worst pain imaginable) at least twice a day, once in the morning and the second time in the afternoon or evening, with the aim of optimizing analgesic therapy. Both oral and intravenous medications are available for this purpose, also in addition to continuous regional anesthesia procedures. Data on complications after surgery and prescriptions for pain medications were obtained from German administrative health care claims data provided by the statutory medical health insurer BARMER. With approximately 9 million policyholders, BARMER is the second largest public German health insurance covering all aspects of medical care (i.e., acute hospital care based on a diagnosis-related group system, outpatient medical care and therapy, medications, remedies, and aids). The case load calculation for the present study (n = 600) was based on the expected case entries in the net-ra registry (annual average of the centers with their own study personnel, calculated for a planned inclusion period of 6 months), the expected percentage of BARMER-insured persons (15%), the percentage of expected consent to participate in the study (66%), and the incidence of moderate to severe postoperative chronic pain described for Europe six months after surgery (16%) [11]. As the corona pandemic led to a drastic reduction in elective surgeries, the inclusion period was extended to a maximum of 13 months.

## Study population and data handling

During the maximised inclusion period from March 1, 2021 to March 31, 2022, adult patients from 11 German hospitals scheduled for inpatient surgery were recruited. Only patients who were insured with BARMER, for whom a complex pain therapy procedure and a minimum length of stay of three days were planned, were asked to participate in the study and provide informed consent.

The raw data of the study participants were extracted from the net-ra registry, pre-processed by the study leaders at Saarland University Hospital and forwarded to Jena University Hospital in pseudonymized form using the net-ra ID. Participating hospitals provided the BARMER with identifying information (patient name and insurance ID) along with the net-ra-ID. BARMER extracted the claims data required for the study, which covered a 12-month period before and a 6-month period after the index stay. The index hospital stay (day of admission to discharge) was determined based on the date of surgery for the index procedure. Data were pseudonymized using net-ra-ID and provided to Jena University Hospital for linkage and further analyses. Linkage of both data sources was conducted using the net-ra-ID and validated by comparing patient demographics and dates of the index hospital stay between both data sources.

## Exposures

The net-ra registry data were used to define the adequacy of postoperative pain therapy in three alternative ways.

First, we considered patients to have inadequately controlled postoperative pain if the NRS scores with movement were above three at least once on days 1–3 after surgery according to the cut-off for moderate to severe pain described by Gerbershagen [12]. As we consider the ability to move appropriately after surgery including the ability to cough adequately to be very important for recovery, we used the severity of pain with movement as a quality marker. The observation period of three days after surgery corresponded to the planned minimum length of stay of the included patients. Patients were considered to have adequately controled pain if their pain scores with movement were less than four on all three days. Patients with missing pain values were excluded from the analysis (listwise deletion).

Second, pain peaks with an NRS > 6 at least once in the first three postoperative days were used as an alternative indicator of inadequate pain therapy. Patients were considered to have no pain peaks if the NRS values with movement were less than seven on all three days. Patients with missing pain values were excluded from the analysis (listwise deletion).

Third, we calculated the time to sustained adequacy of postoperative pain control in hours for each individual patient, defined by stable low NRS scores of <4 with movement. For this variable, the entire treatment period of the acute pain service was considered. The groups were divided into "fast pain recovery" and "slow pain recovery" using a median split.

## Outcomes

Outcomes were assessed using the claims data. The primary outcome was a composite of complications that occurred in the postoperative course during hospitalization, but at the earliest after completion of the acute pain service. Cardiac, pulmonary, infectious, thromboembolic, and surgical complications were assessed based on inpatient International Classification of Diseases (ICD) codes (S1 Table). We focused solely on postoperative complications coded as secondary diagnoses. This coding is assigned to diagnoses that arise during hospitalization, while primary diagnoses typically represent the reasons for hospitalization. The occurrence of complications was recorded dichotomously (yes/no), whereby for each case, it was analysed whether at least one of the above-mentioned complications had occurred during the postoperative course.

The secondary outcome was prolonged postoperative analgesic use, defined as at least one prescription of an opioid and/or nonopioid and/or co-analgesic during the first 90 days after discharge and at least one additional prescription of these drug groups between days 91 and 180. Our definition of prolonged postoperative analgesic use aligns with the established definition of prolonged postoperative opioid use [13] and has already been used in a similar way [14]. The prescriptions were identified using outpatient Anatomical Therapeutic Chemical Classification (ATC) codes and the drug groups were categorised as follows: opioids (N02A with the exception of codeine, methadone, and levomethadone, which are generally not used for pain therapy in Germany), non-opioids (N02B, M01A, M01B), and co-analgesics (N02BG10, N03AF01/02, N03AX09/12/16, N06AA09/12, N06AX11/16/21). Since chronic postoperative pain is not represented in the German ICD 10, and the diagnosis of chronic pain or chronic pain syndrome is often made after a considerable delay, we assumed that postoperative analgesic use in two consecutive quarters after discharge is a suitable surrogate parameter for chronic postoperative pain.

## Patient characteristics and covariates

Descriptive variables for patient and procedure characteristics, as well as potential confounding factors for each outcome, were determined a priori from all available net-ra and BARMER variables. Only covariates that preceded exposure in time were considered potential confounders to avoid endogenous selection bias (i.e., collider bias) [15]. From the net-ra registry, we used sex, age, height, weight, American Society of Anesthesiologists (ASA) physical status classification score, presence of chronic pain syndrome, surgical procedure code (OPS), pain management procedure, and postoperative

NRS pain scores documented during the rounds of the acute pain service. We performed plausibility checks for sex (male designation excluding obstetrics), age range: 18–100 years, height range: 30–249 cm, weight range: 1–249 kg, and body mass index (BMI) range: 12–85 kg/m². Based on the surgical procedure codes, we assigned the cases to six surgical groups: endoprosthetic, bone, major open abdominal, thoracic, laparoscopic, and others. Information on the preoperative use of various analgesics was extracted from the claims data (binary yes/no). Opioid-naïve patients were defined as those who had not received an opioid prescription in the two subsequent quarters prior to the index hospitalization. This definition was applied analogously to patients who were naïve to non-opioids and co-analgesics. The classification of drug groups was based on the definition of the outcome variable.

## Statistical analysis

Descriptive statistics were first performed to characterize the population, methods used for acute pain therapy, and postoperative course. Nominal and ordinal variables are shown with absolute and relative frequencies and interval-scaled variables with mean values and standard deviations. Potential group differences were tested using Fisher's Exact Test, Pearson's Chi-squared test, or Wilcoxon rank-sum test, depending on the level of measurement of the observed variable.

Logistic regression analysis was used to examine the effects of appropriate pain control on the risk of complications (primary outcome) and prolonged postoperative analgesic use (secondary outcome). As the three exposure variables were highly interdependent, separate regression models were calculated for each variable. Cases that could not be categorized into the respective groups because of missing pain values were excluded from the analysis (listwise deletion). From all available variables, pre-treatment factors relevant from a medical point of view were selected for both exposures and were included as covariates in the regression models. All models were adjusted for sex, age, ASA physical status classification score, BMI, and surgical group. For the models of prolonged postoperative analgesic use, pre-existing chronic pain syndrome at hospital admission and the use of opioids, non-opioids, and co-analgesics in two consecutive quarters before hospital admission were also taken into account. We report unadjusted and adjusted odds ratios for the different exposure variables for both outcomes. To make the effects of the three exposure variables more intuitively comparable, we derived the average marginal effects (AME) from the fitted logistic regression models [16–18]. The AME is the adjusted probability difference $p_1 - p_0$ between the groups $X = 1$ (exposure present) and $X = 0$ (exposure not present) under statistical control of the covariates in the model, presented with a 95% confidence interval and statistical significance.

Statistical analyses were conducted with R (version 4.4.0) using the mfx package to calculate the average marginal effects. A two-sided p-value <0.05 was set as the level of significance.

## Results

Within the specified period, 571 patients were included in this study. Health insurance data were provided for 554 cases, and correct linkage with the net-ra registry data was possible in 539 cases. The details of the case-selection process are shown in Fig 1.

Table 1 shows the demographic and procedural data of our study cohort. We were able to classify 147 patients (27%) as having adequately controlled pain, even with movement. 16 cases (3%) had to be excluded due to incomplete pain data. A total of 376 patients (70%) had inadequately controlled pain according to our definition and suffered, at least temporarily, from relevant pain with movement. Almost one third of these patients experienced pain peaks with an NRS score of 7 or more at least once, and the average time to achieve sustained adequate pain relief was three and a half days.

In our study cohort, patients with inadequate pain control were more likely to experience at least one postoperative complication (in 21% of cases) than those with adequate pain control (12%, P = 0.024). Pulmonary and surgical complications were predominant in both groups (Table 2). Adjusted for sex, age, BMI, ASA physical status classification score, and type of surgery, the odds of having at least one complication during the postoperative course in hospital were more than twice as high for patients with inadequately controlled pain (adjOR 2.56; 95% CI 1.43 to 4.80, P = 0.002).

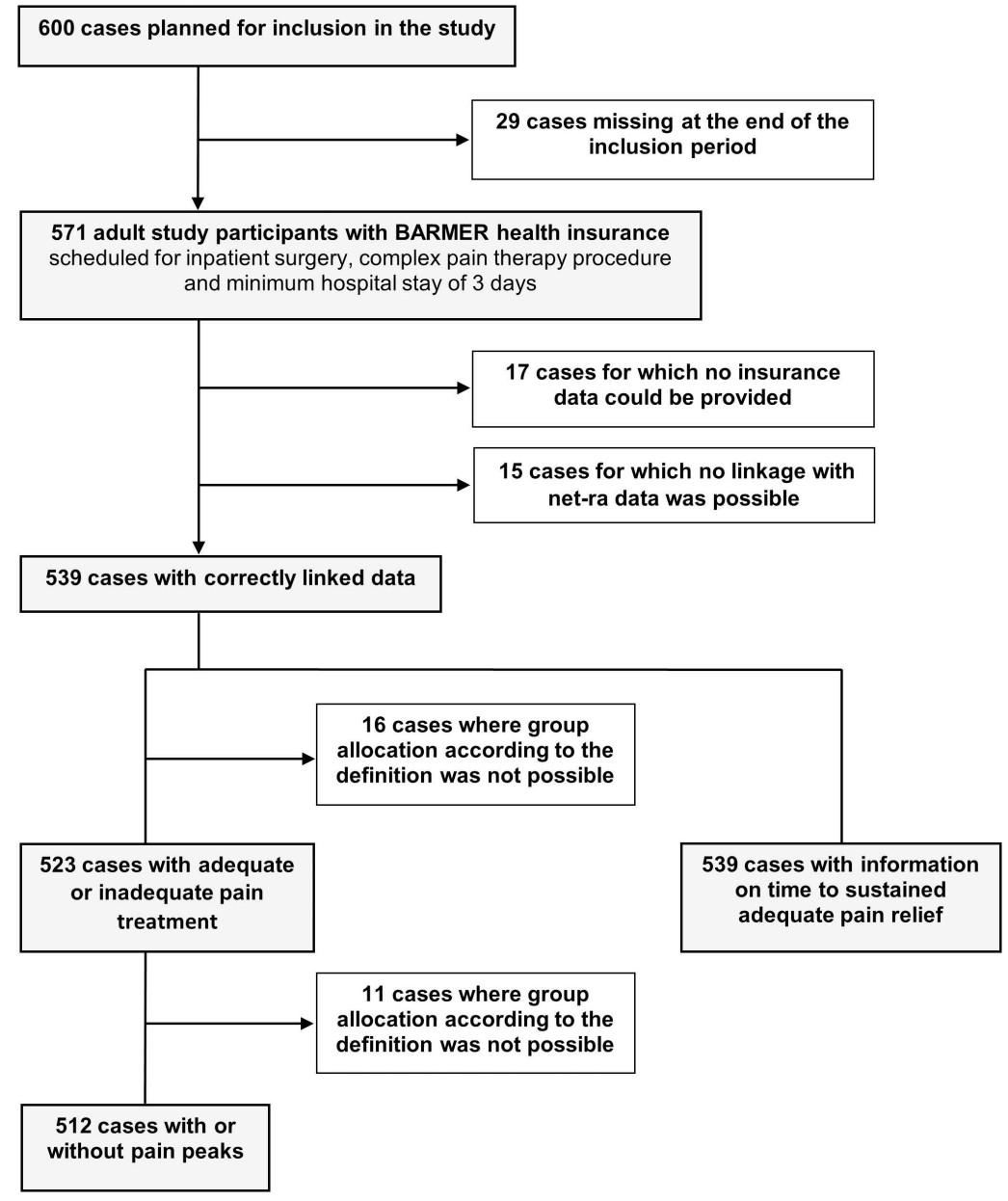

**Fig 1. Flowchart of data selection.**

101 patients (19%) had at least one pain peak of >6 on the NRS scale. According to our definition of pain peaks, 27 cases (5%) had to be excluded due to incomplete pain data. 411 patients (76%) could be assigned to the group without pain peaks according to the definition. More than half of the patients suffering from pain peaks underwent endoprosthetic procedures (56%) and single shot regional anesthesia was the most common analgesic procedure (55%). Pain medication prior to hospitalization was significantly more common than in the group without pain peaks (non-opioids 56% vs. 38%, $P = 0.001$, opioids 29% vs. 14%, $P = 0.001$, S2 Table). However, there was no statistically significant difference between these groups, neither in the frequencies of postoperative complications (16 vs. 18%,

**Table 1. Characteristics of patients with adequately or inadequately controlled postoperative pain.**

| | Adequately controlled postoperative pain $n=147$ | Inadequately controlled postoperative pain $n=376$ | *P* value |
|---|---|---|---|
| **Sex** (male) | 54 (37) | 136 (36) | 0.904 |
| **Age** (years) | 66±15 | 66±14 | 0.612 |
| **ASA physical status classification** | | | 0.502 |
| I | 5 (3) | 21 (6) | |
| II | 72 (49) | 200 (53) | |
| III | 67 (46) | 147 (39) | |
| IV | 3 (2) | 8 (2) | |
| **BMI** (kg m$^{-2}$) | 28±6 | 28±6 | 0.584 |
| **Chronic pain syndrome** | 11 (8) | 35 (9) | 0.508 |
| **Pain medication within 180 days prior to hospitalisation** | | | |
| opioids | 14 (10) | 75 (20) | 0.004 |
| non-opioids | 47 (32) | 170 (45) | 0.006 |
| co-analgesics | 8 (5) | 36 (10) | 0.126 |
| **Type of surgery** | | | <0.001 |
| endoprosthetic | 28 (19) | 159 (42) | |
| bone | 31 (21) | 78 (21) | |
| major general | 41 (28) | 82 (22) | |
| thoracic | 4 (3) | 22 (6) | |
| laparoscopic | 13 (9) | 18 (5) | |
| others | 30 (20) | 17 (5) | |
| **Acute pain treatment** | | | 0.018 |
| patient controlled intravenous | 7 (5) | 43 (11) | |
| regional anesthesia, continuous | 86 (59) | 174 (46) | |
| regional anesthesia, single shot | 53 (36) | 151 (49) | |
| others | 1 (0.7) | 8 (2) | |
| **Time to sustained adequate pain relief** (hours) | – | 83±51 | |
| **Presence of at least one pain peak NRS > 6 during postoperative days 1–3** | 0 (0) | 101 (27) | <0.001 |

Data are presented as mean±SD or *n* (%). Pain was defined as adequately controlled if NRS scores on movement were consistently <4 during postoperative days 1–3, and inadequately controlled if there was at least one NRS pain score on movement ≥4 within postoperative days 1–3. According to this definition, 16 cases (3%) had to be excluded due to incomplete pain data.

p = 0.571, S3 Table) nor in the logistic regression model with covariate adjustment (adjOR 1.27; 95% CI 0.64 to 2.42, *P* = 0.478).

337 (63%) patients with fast and 202 (37%) patients with slow pain recovery showed no clinically meaningful demographic differences (S4 Table). The most common surgery was endoprosthetic surgery, in the slow pain recovery group followed by major general surgery (32 vs. 18%), and in the fast pain recovery group followed by bone surgery (26 vs. 14%). Almost two-thirds of the patients in the slow pain recovery group were treated with continuous regional anesthesia, compared to almost half of the patients in the fast pain recovery group. Pain peaks with NRS values >6 occurred more frequently in patients who recovered slowly (27 vs. 16%, *P* = 0.002). The number of patients with at least one postoperative complication was twice as high among these patients (27 vs. 13%, *P* = 0.001, S5 Table). After adjusting for sex, age,

**Table 2. Postoperative complications in patients with adequately or inadequately controlled postoperative pain.**

| | Adequately controlled postoperative pain $n=147$ | Inadequately controlled postoperative pain $n=376$ | P value |
|---|---|---|---|
| **Inpatient complications** | | | |
| cardiac | 1 (0.7) | 1 (0.3) | 0.484 |
| pulmonary | 11 (8) | 47 (13) | 0.101 |
| infectious | 3 (2) | 15 (4) | 0.272 |
| thromboembolic | 1 (0.7) | 13 (4) | 0.127 |
| surgical | 5 (3) | 34 (9) | 0.027 |
| Composite: at least one of the above complications | 18 (12) | 78 (21) | **0.024** |
| **Postoperative use of analgesics for at least 6 months** | | | |
| opioids | 3 (2) | 25 (7) | 0.035 |
| non-opioids | 15 (10) | 57 (15) | 0.139 |
| co-analgesics | 3 (2) | 15 (4) | 0.272 |
| Composite: any of the above | 16 (11) | 77 (21) | **0.010** |

Data are presented as $n$ (%). Pain was defined as adequately controlled if NRS scores on movement were consistently <4 during postoperative days 1–3, and inadequately controlled if there was at least one NRS pain score on movement ≥4 within postoperative days 1–3.

BMI, ASA physical status classification score, and type of surgery, patients with slow recovery from postoperative pain had more than double the odds of at least one complication during hospitalization (adjOR 2.21; 95% CI 1.35 to 3.64, $P=0.002$).

When comparing patients with adequately and inadequately controlled pain in terms of prolonged postoperative use of analgesics, there was no statistically significant difference in the odds (adjOR 1.87; 95% CI 0.98 to 3.72, $P=0.064$; adjusted for sex, age, BMI, ASA physical status classification score, diagnosis of chronic pain syndrome at the time of hospital admission, opioid or nonopioid or co-analgesics medication within 6 months before surgery, and type of surgery). The comparison of the groups with and without pain peaks revealed no significant difference in the adjusted odds (adjOR 1.20; 95% CI 0.65 to 2.19, $P=0.555$). Multiple logistic regression analysis of patients with fast or slow pain recovery, which was also adjusted for the above-mentioned covariates, showed comparable odds for prolonged postoperative analgesic use (adjOR 0.95; 95% CI 0.55 to 1.60, $P=0.836$). We present a summary of the results and detailed results of the individual models in the supplemental online content (S6 and S7 Tables).

To better compare the effect sizes of the three selected potential influencing factors on the probability of postoperative complications and prolonged analgesic consumption, we also calculated the average marginal effects with 95% confidence intervals, controlling for the above-mentioned confounders. The probability of a complication occurring during the hospital stay was increased by 12% if postoperative pain was inadequately controlled (NRS > 3) in the first three days after surgery (0.12; 95% CI 0.05 to 0.20; $P=0.002$), although an influence of pain peaks could not be detected (0.03; 95% CI −0.05 to 0.12; $P=0.478$). Patients who were slow to recover from postoperative acute pain were 10% more likely to have complications (0.10; 95% CI 0.04 to 0.16; $P=0.001$; Fig 2). The strongest signal for a higher probability of prolonged postoperative dependence on analgesics was due to inadequately controlled pain, although this was not statistically significant (0.07; 95% CI 0.00 to 0.15; $P=0.062$; Fig 3). Pain peaks during the first three postoperative days and the speed of pain recovery had no influence on the occurrence of prolonged postoperative analgesic use in our analysis (pain peaks: 0.02; 95% CI −0.05 to 0.10; $P=0.555$; pain recovery: −0.01; 95% CI −0.07 to 0.06; $P=0.836$).

**Average marginal effects** (95% CI) for **inpatient complications after surgery**

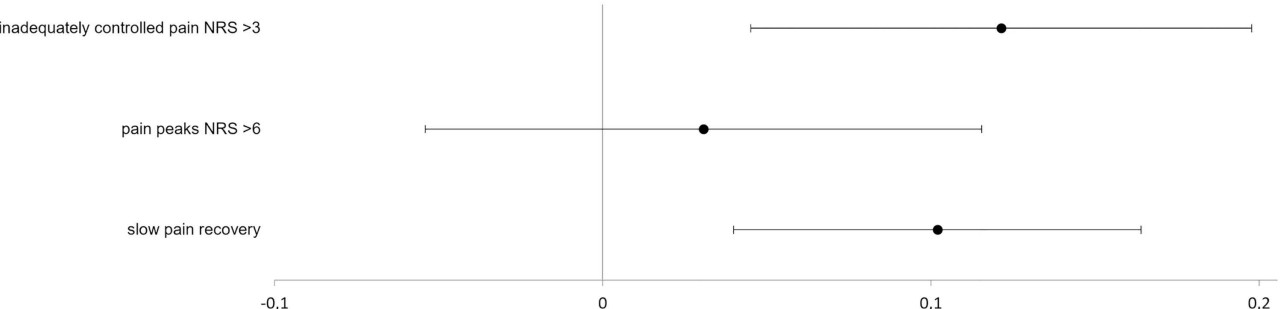

**Fig 2. Influence of quality indicators of pain therapy on the occurrence of postoperative inpatient complications.** Inadequately controlled pain was defined as pain with movement NRS > 3 at least once within postoperative days 1-3, events with an NRS > 6 were considered as pain peaks, and slow pain recovery was defined as a time above the median to reach sustained pain scores below 4 (11-point NRS) *Average marginal effects controlled for age, sex, American Society of Anesthesiologists physical status classification score, Body Mass Index, and groups of surgery.

**Average marginal effects** (95% CI) for **prolonged postoperative use of analgesics**

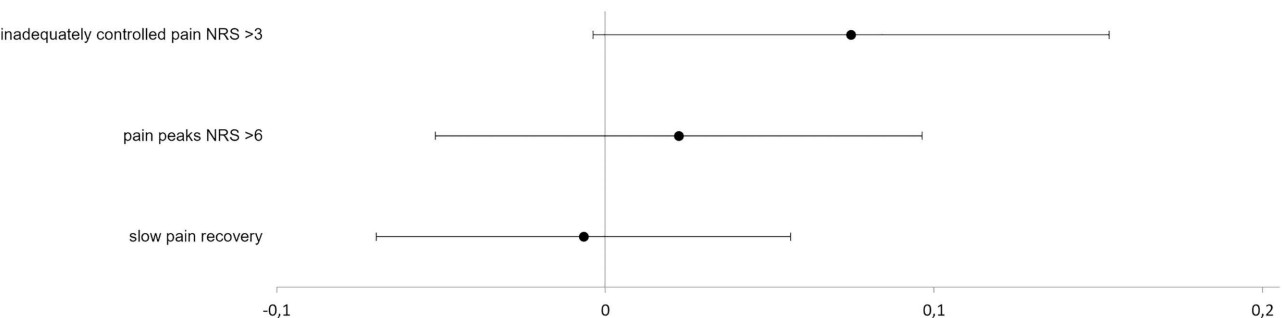

**Fig 3. Influence of quality indicators of pain therapy on the occurrence of prolonged analgesic use.** Inadequately controlled pain was defined as pain with movement NRS > 3 at least once within postoperative days 1-3, events with an NRS > 6 were considered as pain peaks, and slow pain recovery was defined as a time above the median to reach sustained pain scores below 4 (11-point NRS) *Average marginal effects controlled for age, sex, American Society of Anesthesiologists physical status classification score, Body Mass Index, preoperative opioid use, preoperative non-opioid use, preoperative co-analgesic use, and groups of surgery.

## Discussion

As expected, patients whose pain was inadequately controlled in the first three days after surgery were more than twice as likely to experience postoperative complications than patients whose pain was adequately controlled. The occurrence of pain peaks showed no significant influence. Patients who slowly recovered from postoperative pain were twice as likely to experience complications as patients who had a rapid pain recovery. This findings support our hypothesis that adequate postoperative pain therapy reduces the risk of such adverse events which is consistent with the results of a few studies available on this issue. Turan et al. showed that each one-point increase in an 11-point pain score increased the adjusted odds of postoperative myocardial injury by 22% in patients undergoing non-cardiac surgery [6]. Van Helden et al. studied

patients after major abdominal surgery and found that the mean pain score on the first postoperative day was a significant predictor of infectious and non-infectious 30-day postoperative complications (OR=1.116, 95% CI 1.053 to 1.183) arising after the first postoperative day [7]. In a broad variety of surgical procedures analyzed by van Boekel et al., patients who perceived their pain as unacceptable had twice the risk of complications [8], which is exactly in line with our findings for inadequately controlled pain.

There is some evidence that the burden of postoperative pain is a predictor for persistent postsurgical pain. A European observational study identified the percentage of time in severe pain at postoperative day one as an independent risk factor for the incidence of chronic postsurgical pain [11]. For breast surgery, a systematic review and meta-analysis showed 16 percent greater odds for persistent postsurgical pain with every one-point-increase on an 11-point visual analogue scale [19]. Higher postoperative acute pain following inguinal hernia repair surgery was also strongly associated with persistent postsurgical pain after 12 months [20], and patients with a higher severity of pain during the first three days after thoracotomy or -scopy had a higher likelihood of developing chronic pain related to thoracic surgery at 6 months [21]. Since chronic postoperative pain is not represented in the German ICD 10, we used prolonged postoperative use of analgesics as a potential indicator. Although there was a clear signal that inadequately controlled pain could increase the risk of prolonged postoperative analgesic use, no statistically significant effect of the above-mentioned pain therapy characteristics was found. However, in our small population this signal may have been too weak to reach statistical significance in view of the strong effects of pre-existing chronic pain syndrome and the preoperative use of analgesics.

The importance of pain peaks is unclear. Associations between episodes of severe pain during the first 72 hours after surgery and persistent postsurgical pain were found after thoracotomy [22], and breast surgery [23]. In our analysis, however, we were unable to determine any significant influence of pain peaks on the occurrence of postoperative complications or prolonged postoperative analgesic use, possibly due to low case and event numbers. Further research should clarify whether complication risks actually increase with pain intensity, or whether exceeding a certain threshold intensity is a major driver.

The long-term use of analgesics is particularly important when opioids are involved. Dreiling et al. estimated the overall rate of long-term postoperative opioid use in Germany to be 1.4% [95% CI 1.4–1.5%] based on a sample of 203,327 opioid-naive patients who underwent surgery [24]. This rate appears to be significantly lower than in other countries which may be due to the fact that the dispensing of pain medication to patients is handled very restrictively. For example, when patients are discharged, postoperative opioids are only prescribed on a special prescription form indicating the lowest possible quantity and a validity period of only 3 days. Non-opioids are only available over the counter in Germany in low doses and limited quantities. We consider the exclusive use of prescription non-opioids in direct temporal association with surgery to be robust in our analysis.

The chosen cut-off for moderate to severe pain [12] seems to be a good threshold for pain-triggered stress reactions and is widely used to distinguish irrelevant from relevant pain [8,25–28].

In our study, pain was considered inadequatly controlled if pain scores exceeded a NRS score of three at any point during the first three postoperative days. This strict definition was deemed necessary in relation to our research question to clearly distinguish between patients who did not experience relevant pain and those who did. Multidimensional pain assessments using a biopsychosocial pain model describe pain in a more holistic way. However, the basis for using this model in the acute postoperative situation is less well established and subject of current debate [29]. Furthermore, it is not yet part of the routine documentation of acute pain services.

Considering the individual time that a patient needs to overcome postoperative pain, which means having stable, at most mild pain levels, is a new approach. In a previous net-ra analysis, we showed that in patients with chronic pain, postoperative acute pain took longer to resolve [30]. In the present analysis, we used individual time to reach sustained mild pain levels as a possible predictor for postoperative outcomes. The distinction between slow and fast pain recovery using a median split was suitable to show significant effects on the occurrence of postoperative complications, but not on

prolonged use of analgesics. Analyses of pain trajectories also take this time factor into account revealing associations between poor recovery from postoperative pain and the occurrence of postoperative chronic pain [31]. Current research suggests that both the presence of moderate to severe pain and the time needed to achieve adequate pain relief have an impact on postoperative outcomes. Further research should attempt to clarify how these two components are related and how, for example, a "critical pain burden" could be determined.

Individual patient factors such as genetic predisposition, comorbidities, medication, or psychological co-factors appear to play an important role [30–33]. Further research is needed to screen patients with an increased sensitivity to pain and to offer them the most effective pain therapy possible.

## Limitations

BARMER health data is regularly used for scientific analysis and contains reliable inpatient diagnoses with regular sample validation of internal and external validity [34,35]. Approximately 90 percent of the German population has statutory health insurance, with BARMER, the second-largest statutory health insurance fund, covering around 9 million people (12%). Compared to other statutory health insurance companies, its beneficaries are on average more likely to be female and younger for what we accounted in our statistical models. Privately insured individuals, who make up approximately 10% of the German population and differ in many ways from those with statutory insurance [36], were not represented in our analysis. As a consequence, generalizability of the results is potentially limited as the representativeness of the sample with respect of the target population is not ensured. Our study included adult inpatients whose planned surgery was expected to result in severe postoperative pain and who therefore received comprehensive postoperative pain management by qualified staff. It can be assumed that the case severity was higher, particularly compared to outpatient surgery, so the results are not fully transferable.

Health insurance data is coded to bill the health insurance company as cost-effectively as possible. This can lead to scientifically relevant but not billing-relevant issues not being coded and billing-relevant issues being coded rather generously. However, it is not to be expected that this bias will differ within the groups that we formed based on the pain values.

We suspect that adequate pain control after major surgery has an influence on the risk of postoperative complications. However, drawing causal conclusions from observational studies relies on certain assumptions, as outlined in the Neyman-Rubin causal model (NRCM) [37,38]. The Average Marginal Effect (AME), derived from the logistic regression model, serves as an unbiased estimator of the Average Treatment Effect (ATE) under the conditions that (a) all confounding variables are properly accounted for (i.e., strong ignorability and correct model specification) and that these covariates are accurately measured. However, due to the study's design, we cannot completely eliminate the possibility of unmeasured confounders, which can introduce residual bias in the adjusted odds ratio and AME.

We used listwise deletion for patients with incomplete records (e.g., incomplete pain scores over time). Unbiased parameter estimation using listwise deletion requires that the missing data mechanism is missing completely at random, which cannot be ensured in our study. However, since no more than five percent of cases were affected in any analysis, no relevant bias can be assumed [39].

## Conclusions

We found empirical evidence supporting our hypothesis that the quality of postoperative control affects the occurrence of postoperative complications. Additionally, we observed a trend toward prolonged analgesic use, which serves as a surrogate indicator for the development of chronic pain. Postoperative acute pain should, therefore, be consistently assessed until resolvement, and all treating specialists should strive to minimize pain load. Further research into patient- and procedure-specific factors and individual treatment options is important in order to reduce the undesirable consequences of pain.

## Supporting information

**S1 Table. International Classification of Diseases (ICD) Codes used for the implementation of postoperative inpatient complications.**
(DOCX)

**S2 Table. Characteristics of patients with and without pain peaks (NRS > 6 with movement) within the first three postoperative days.** 27 cases (5%) had to be excluded due to incomplete pain data. Values are number (proportion) or mean (standard deviation) as appropriate.
(DOCX)

**S3 Table. Postoperative complications in patients with and without pain peaks (NRS > 6 with movement) within the first three postoperative days.** Values are numbers and proportions.
(DOCX)

**S4 Table. Characteristics of patients with fast and slow pain recovery.** Rapid pain recovery was defined as a time to reach sustained pain scores below 4 (NRS) with movement below the median time of the group. Times above the median were classified as slow pain recovery. Values are number (proportion) or mean (standard deviation) as appropriate.
(DOCX)

**S5 Table. Postoperative complications in patients with rapid or slow pain recovery.** Rapid pain recovery was defined as a time to reach sustained pain scores below 4 (NRS) with movement below the median time of the group. Times above the median were classified as slow pain recovery. Values are numbers and proportions.
(DOCX)

**S6 Table. Crude and adjusted Odds ratios (OR), and average marginal effects (AME) for postoperative complications and prolonged postoperative use of analgesics for patients with inadequately controlled postoperative pain, pain peak experience, or slow pain recovery.** Inadequately controlled pain was defined as NRS values with movement ≥4 at least once within postoperative days 1–3, pain peaks as NRS values >6 with movement at least once within the first three postoperative days, and slow pain recovery as a time to reach sustained pain scores below 4 (NRS) with movement above the median time of the study population. [a]Adjusted for sex, age, BMI, ASA score, and type of surgery. [b]Adjusted for sex, age, BMI, ASA score, diagnosis of chronic pain syndrome at the time of hospital admission, opioid medication within 6 months before surgery, nonopioid medication within 6 months before surgery, co-analgesics medication within 6 months before surgery, and type of surgery.
(DOCX)

**S7 Tables. Detailed results of the different binary logistic regression models (a-f).**
(DOCX)

## Acknowledgments

We would like to thank all hospital centers that participated in the LOPSTER project and provided data for the current analysis:

Allgemeines Krankenhaus Celle, Prof. Dr. med. Dieter Fröhlich;
Christliches Klinikum Unna West, Dr. med. Wolf Armbruster;
DIAKOVERE Friederikenstift Hannover, Prof. Dr. med. André Gottschalk;
DIAKOVERE Henriettenstift Hannover, Prof. Dr. med. André Gottschalk;
Josephs-Hospital Warendorf, Dr. med. Alexander Reich;
Kliniken Calw, Klinikverbund Südwest, Dr. med. Jens Döffert;

Marienhospital Stuttgart, Prof. Dr. med. René Schmidt;

Ubbo-Emmius-Klinik gGmbH, PD Dr. med. Markus Paxian;

Universitätsklinikum des Saarlandes Homburg/Saar, Univ.-Prof. Dr. med. Thomas Volk; Universitätsklinikum Marburg (UKGM), Univ.-Prof. Dr. med. Hinnerk F. W. Wulf;

Universitäts- und Rehabilitationskliniken Ulm (RKU), Dr. med. Jörg Winckelmann

For further information visit www.net-ra.eu

We would also like to thank Jana Schmitt for the detailed organisation of data acquisition and support of the participating hospitals, and Christin Arnold for managing the project over the entire term.

## Author contributions

**Conceptualization:** Christine Kubulus, Marcus Komann, Markus Paxian, Ann-Kristin Schubert, Daniel Schwarzkopf, Ursula Marschall, Johannes Dreiling, Thomas Volk.

**Data curation:** Marcus Komann, Norman Rose, Johannes Dreiling, Carolin Fleischmann-Struzek.

**Formal analysis:** Marcus Komann.

**Funding acquisition:** Daniel Schwarzkopf, Ursula Marschall, Thomas Volk.

**Investigation:** Christine Kubulus, Winfried Meissner, Thomas Volk.

**Methodology:** Christine Kubulus, Marcus Komann, Daniel Schwarzkopf, Norman Rose, Ursula Marschall, Johannes Dreiling, Carolin Fleischmann-Struzek, Thomas Volk.

**Project administration:** Christine Kubulus.

**Resources:** Norman Rose, Thomas Volk.

**Supervision:** Christine Kubulus, Daniel Schwarzkopf, Winfried Meissner, Thomas Volk.

**Validation:** Christine Kubulus, Marcus Komann, Markus Paxian, Ann-Kristin Schubert, Norman Rose, Thomas Volk.

**Writing – original draft:** Christine Kubulus.

**Writing – review & editing:** Marcus Komann, Markus Paxian, Ann-Kristin Schubert, Daniel Schwarzkopf, Norman Rose, Winfried Meissner, Ursula Marschall, Johannes Dreiling, Carolin Fleischmann-Struzek, Thomas Volk.

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
