## [Decision Letter · Decision Letter 0]

25 Jun 2025

PONE-D-25-25651Quality of pain release after major surgery and its association with postoperative complications – a prospective observational studyPLOS ONE

Dear Dr. Kubulus,

Thank you for submitting your manuscript to PLOS ONE. After careful consideration, we feel that it has merit but does not fully meet PLOS ONE’s publication criteria as it currently stands. Therefore, we invite you to submit a revised version of the manuscript that addresses the points raised during the review process.

Thank you for submitting your manuscript. Your work is very interesting, but there are some points that need to be revised or clarified. Please address all points of criticism from the reviewers.==============================

We look forward to receiving your revised manuscript.

Kind regards,

Alexander Wolf

Academic Editor

PLOS ONE

 [The LOPSTER project on which this publication is based was funded by the German Innovation Committee of the Federal Joint Committee (G-BA), Berlin, Germany under grant no. 01VSF19019 to Daniel Sschwarzkopf.]. 

3. In the online submission form, you indicated that [Data are available from the authors upon reasonable request and after approval by the net-ra registry’s organizational committee and by the responsible data protection officer. Data can only be provided in an anonymous form if complete anonymisation is possible.].

Additional Editor Comments (if provided):

Reviewers' comments:

Reviewer's Responses to Questions

**Comments to the Author**

1. Is the manuscript technically sound, and do the data support the conclusions?

Reviewer #1: Yes

Reviewer #2: Yes

2. Has the statistical analysis been performed appropriately and rigorously? 

Reviewer #1: Yes

Reviewer #2: Yes

3. Have the authors made all data underlying the findings in their manuscript fully available?

Reviewer #1: No

Reviewer #2: No

4. Is the manuscript presented in an intelligible fashion and written in standard English?

Reviewer #1: Yes

Reviewer #2: Yes

5. Review Comments to the Author

Reviewer #1: Dear authors,

This is a well-conducted prospective observational study evaluating whether poor postoperative pain control is associated with higher rates of complications and prolonged analgesic use. Very nice initiative while research on this topic remains challenging.

In the age of ever-increasing knowledge regarding perioperative safety, I appreciate the opportunity to carefully read and comment regarding the submission “Quality of pain release after major surgery and its association with postoperative complications – a prospective observational study”. I really like the idea of studying. I congratulate the authors for the large amount of analyzed data. Overall, the study is well reported and complete.

Major comments

- Although the study is observational, the language in some parts (e.g., “postoperative pain causes complications”) could be misinterpreted. Clarify that findings show associations, not causation. From my early view, I feel the aim (and ambition) here is an explanation and causal inference. Let us say, ¿is poorly controlled postop pain causing postop. complications? However, during my lecture of the manuscript I have seen only a few details about causal interpretation of the calculated estimates with some attempts to avoid the word “cause.” Moreover, the analytic intentions were focused on association which is fine but the interpretation -in some aspects- is causal and that should be clarified. I consider the interpretation of the estimates (and confidence intervals) could be improved -and there are plenty of recent literature to help this way- (i.e. doi:10.2105/AJPH.2018.304337). Are you sure you just want to say, “associated with”. Following this line, the main abstract conclusion is causal.

- I suggest you should state why your intentions are not directly causal (if it is what you were aiming for). Target trial strategy where a causal analysis in observational data can be viewed as an attempt to emulate a hypothetical trial—the target trial could be useful (doi: 10.1093/aje/kwv254).

- Considering you propose the use of the average marginal effects, I really miss the reporting of a causal framework (AME is used as a proxy of the ATE). I cannot understand why you did not consider the usefulness of drawing your assumptions before making conclusions (and stay only as an association having the data and the scientific ambition). May you consider it useful to draw a DAG regarding this potential causal relationship -if you condeir this-. Please consider adding a DAG to directly address 1) a causal pathway, 2) a mediator, 3) potential confounders, and/or 4) colliders in the causal structure. Do you consider if some of the included adjusted variables in your models could be a collider and not a confounder ? If you are only considering association, well, that is fine but should clarify and improve your conclusions.

- The reported complications, (lines 260-261 “In patients with inadequate pain control, at least one postoperative complication occurred more frequently (22%) than in the adequately treated group (16%).”) should be reported besides their CI95% bc those are direct estimates of risk.

- The definition of “prolonged analgesic use” as a proxy for chronic pain is reasonable but could benefit from validation or at least a more detailed justification, given the lack of a diagnostic ICD code.

- The exclusion of patients with incomplete pain data could introduce bias. Authors should provide a sensitivity analysis or at least a discussion of this potential issue. There is no major information about the management of missing data.

Pain scores are patient-reported but also dependent on measurement frequency and method, which may vary between institutions. Addressing potential inter-center variability in pain assessment could influence those reporting. Please may you discuss something about.

- Discussion does not mention at all any intent to interpret the result as causal. There are four (or five) major assumptions of causality in cohort studies and big data. Exchangeability, Positivity, Consistency, Temporality and Target population. Please address each one in your discussion in order to assess if your study fits some of them (if any) and their interpretation with a causal approach.

Minor comments

Abstract.

The study addresses an important and underexplored association between early postoperative pain control and surgical outcomes. I may say the abstract is not well written and perhaps incomplete. Methods section of the abstract should be improved to improve the overall understanding of the process. Add exact sample size (n=539) to enhance transparency.

Introduction:

First two lines (69-72) require a reference and I consider there are plenty of.

Lines 73-74 states you are expanding about association of. Is that the cause or not ? Is there any studies focused on studying -using proper analytical methods- causality in this scenario of much of the literature is discussion association.

Line 76-77 suggests a potential cause of the complications beyond any association.

LInes 78-85. Is that enough for association studies? Why not address the problem of causality more focused on. Or please expand about the analytical nature of those mentioned studies.

I strongly miss the main objective of this study. I mean, both hypotheses sound interesting but please state clearly the aims of this study including if your aiming causation or only association. From the Record declaration you mentioned later on “Because the strengths and limitations of methods used to conduct research with routinely collected data may be contentious, a clear description of a study’s objectives is essential”.

Methodology:

Prospective design with large sample size and linkage to real-world insurance data adds robustness. Please state in line 100 you are using routinely collected data.

I can see you selected high risk patients for postop severe pain (lines 134-136 “Only patients who were insured with BARMER GEK, for whom a complex pain therapy procedure and a minimum length of stay of three days were planned, were asked to participate in the study and provide informed consent.”

Please clarify lines 177-179 “To ensure that the complications mentioned were not the reason for hospitalization but occurred after the surgical procedure, we only considered secondary diagnoses and not principal diagnoses.”.

The exclusion of patients with incomplete pain data could introduce bias. Authors should provide a sensitivity analysis or at least a discussion of this potential issue. There is no major information about the management of missing data.

Results

Figure 2 is informative but dense. Consider separating pain outcomes and complication outcomes for clarity.

Please use consistent terminology throughout (e.g., "chronic postsurgical pain" vs. “prolonged analgesic use”).

Tha main adjusted regression model for the primary outcome under study should be presented in the main text being carefully to avoid any interpretation of other coefficients (which in general is correct and appropriate).

Discussion

First paragraph of the discussion is very clear.

Second paragraph is more useful. I would suggest making a Table regarding the availability of similar studies or studies addressing the same association.

Final conclusion, however includes a causal statement that should be considered “Postoperative acute pain should, therefore, be consistently assessed until

resolvement, and all treating specialists should strive to minimize pain load.”.

Reviewer #2: Review PONE-D-25-25651

Quality of pain release after major surgery and its association with postoperative complications – a prospective observational study

In general:

Thank you for the invitation to review this paper.

In this prospective multicenter observational study the authors tested the hypothesis that postoperative complications occur more frequently in patients with poor pain control regarding both severity and duration. Second, that the likelihood of pain chronification is higher in these patients, reflected by prolonged postoperative analgesic consumption.

The study is well-designed and makes original use of various linked registries. Considering the individual time that a patient needs to overcome postoperative pain, which means having stable, at most mild pain levels, as a new approach is interesting.

In the use of the registries, understandable methodological choices have been made, which may, however, influence the results. I therefore suggest examining these choices carefully and providing a more detailed discussion of potential limitations and points for consideration."

Major comments:

Methods:

The inclusion of only patients insured with BARMER GEK may introduce a risk of selection bias, as membership in this statutory health insurance fund may not be representative of the general German population. Individuals insured by BARMER may differ systematically from those covered by other statutory or private insurers with respect to socioeconomic status, age, health status, or regional distribution. For example, higher-income individuals are more likely to opt for private health insurance, and BARMER may have a higher proportion of older or chronically ill patients compared to smaller or more specialized funds. Therefore, the study findings may have limited generalizability beyond the insured population of BARMER GEK and introducing other risk factors for postoperative pain. I suggest to comment this in the discussion section.

When considering patients to have inadequately controlled postoperative pain if the NRS scores with movement were above three at least once on days 1-3 after surgery (according to the cut-off for moderate to severe pain described by Gerbershagen), this means that patients with an NRS4 once in those three days are considered to have inadequate pain control. Although used in other studies, to my clinical opinion this is far to strict and may have lead to more patients in the inadequate pain control group.

When checking literature, the results of the paper of van Boekel et al.(2017) reinforces my clinical opinion. I would suggest not only stating that the cut-off is widely used, but also discussing the cut-off in the light of personalized levels of pain scores.

van Boekel, R. L. M., Vissers, K. C. P., van der Sande, R., Bronkhorst, E., Lerou, J. G. C., & Steegers, M. A. H. (2017). Moving beyond pain scores: Multidimensional pain assessment is essential for adequate pain management after surgery. PloS one, 12(5), e0177345. https://doi.org/10.1371/journal.pone.0177345

Results:

Mentioned in text of results (page 15, line 265): “In patients with inadequate pain control, at least one postoperative complication occurred more frequently (22%) than in the adequately treated group (16%).” (p-value missing). However, in table 2 it shows that “Composite: at least one of the above complications” is 21% versus 12%? Please explain.

Mentioned in text of results (page 18, line 316): “Pain peaks during the first three postoperative days and the speed of pain recovery had no influence on the occurrence of chronic postoperative pain in our analysis (pain peaks: 0.02; 95% CI -0.05 to 0.10; P=0.555; pain recovery: -0.01; 95% 319 CI -0.07 to 0.06; P=0.836).” I think you mean “prolonged postoperative analgesic use“ as a proxy for chronic postoperative pain, as stated in methods? Please explain.

Discussion:

You were unable to demonstrate any influence of the aforementioned pain therapy characteristics on prolonged postoperative analgesic use. I would suggest to add a paragraph in the discussion to discuss prolonged postoperative analgesic use as a proxy for chronic postoperative pain, in the light oft he German culture of opioid prescribing which may differ from other countries?

Minor:

Abstract:

Please correct objektives in objectives

Please add % to numbers of patients in results, for example page 16, line 270. And p-values when comparing results, for example page 15, line 261.

In results is stated: “were more than twice as high for patients with inadequately controlled pain (adjOR 2.56; 95% CI 1.43 266 to 4.80, P=0.002).” The same results are described in the abstract as: “had almost three times the risk of complications (adjOR 2.56; 48 95% CI 1.43-4.80, p=0.002)”. In my opinion, 2.56 is not almost 3, please adjust.

Table 1, what do you mean with “Definition not applicable” underneath table 1?

6. PLOS authors have the option to publish the peer review history of their article (what does this mean? ). If published, this will include your full peer review and any attached files.

**Do you want your identity to be public for this peer review?** For information about this choice, including consent withdrawal, please see our Privacy Policy .

Reviewer #1: **Yes: ** Jose A Calvache

Reviewer #2: No

---

## [Author Response · Author response to Decision Letter 1]

21 Aug 2025

Many thanks to the academic editor, Dr. Alexander Wolf, for the opportunity to revise our manuscript. We have now complied with the journal's style requirements and provide the necessary information about the funder and data availability in the cover letter.

We would also like to thank the two reviewers for taking the time to read our work carefully and for their very helpful and appropriate comments. Please find our responses to the individual points that were addressed in the separate file "Response to Reviewers".

---

## [Decision Letter · Decision Letter 1]

8 Sep 2025

Does the quality of pain relief after major surgery influence the risk of postoperative complications? A prospective observational study

PONE-D-25-25651R1

Dear Dr. Kubulus,

We’re pleased to inform you that your manuscript has been judged scientifically suitable for publication and will be formally accepted for publication once it meets all outstanding technical requirements.

Kind regards,

Alexander Wolf

Academic Editor

PLOS ONE

Additional Editor Comments (optional):

Reviewer #1:

Reviewer #2:

Reviewers' comments:

Reviewer's Responses to Questions

**Comments to the Author**

1. If the authors have adequately addressed your comments raised in a previous round of review and you feel that this manuscript is now acceptable for publication, you may indicate that here to bypass the “Comments to the Author” section, enter your conflict of interest statement in the “Confidential to Editor” section, and submit your "Accept" recommendation.

Reviewer #1: All comments have been addressed

Reviewer #2: (No Response)

2. Is the manuscript technically sound, and do the data support the conclusions?

Reviewer #1: Yes

Reviewer #2: Yes

3. Has the statistical analysis been performed appropriately and rigorously? 

Reviewer #1: Yes

Reviewer #2: Yes

4. Have the authors made all data underlying the findings in their manuscript fully available?

Reviewer #1: Yes

Reviewer #2: No

5. Is the manuscript presented in an intelligible fashion and written in standard English?

Reviewer #1: Yes

Reviewer #2: Yes

6. Review Comments to the Author

Reviewer #1: I would like to sincerely thank the authors for thoroughly addressing all the comments provided in the previous round. The revisions have significantly strengthened the manuscript. This version presents a nearly comprehensive analysis of an observational study, and it does an excellent job of emphasizing the inherent challenges and pitfalls that arise when interpreting associations as causal relationships. The added clarifications contribute to the overall clarity and rigor of the work, making it much more informative for the readers incluiding th peer review response. Thanks

Reviewer #2: Thank you for the excellent scientific discussion, while answering the comments. I was a pleasure to read your thoughts and suggestions. All my comments are fully addressed.

7. PLOS authors have the option to publish the peer review history of their article (what does this mean? ). If published, this will include your full peer review and any attached files.

**Do you want your identity to be public for this peer review?** For information about this choice, including consent withdrawal, please see our Privacy Policy .

Reviewer #1: **Yes: ** Jose A. Calvache MD MSc PhD

Reviewer #2: No

---

## [Editor Report · Acceptance letter]

PONE-D-25-25651R1

PLOS ONE

Dear Dr. Kubulus,

I'm pleased to inform you that your manuscript has been deemed suitable for publication in PLOS ONE. Congratulations! Your manuscript is now being handed over to our production team.

Kind regards,

on behalf of

Dr. Alexander Wolf

Academic Editor

PLOS ONE